# Hidden Adversarial Patch Attacks for Optical Flow

**Benjamin J. Wortman** [1]

## Abstract

Adversarial patches have been of interest to researchers in recent years due to their easy implementation in real world attacks. In this paper we expand upon previous research by demonstrating a new *hidden* patch attack on optical flow. By altering the transparency during training we can generate patches that are invariant to their background meaning they can be inconspicuously applied using a transparent film to any number of objects. This also has the added benefit of reducing training costs when mass producing adversarial objects, since only one trained patch is needed for any application. Although this specific implementation is demonstrated using a white box attack on optical flow, it can be generalized to other scenarios such as object recognition or semantic segmentation.

## 1. Introduction

As a number of adversarial examples have been demonstrated on photo data in recent years, it's only natural to add another dimension and extend this to video data. Despite the additional complexity associated with adding the temporal dimension, this also provides new attack vectors that can be leveraged by bad actors. One potential attack vector is disrupting optical flow, a popular modality for action recognition and object tracking. Optical flow describes the apparent movement of pixels in an image sequence and is typically represented as a vector field (u,v) that corresponds to the displacement of each pixel in the image sequence.

Adversarial patches attacking optical flow have serious implications for automated driving systems that use deep learning optical flow algorithms as an input. This is a safety critical application, so if an effective attack against these systems can be demonstrated then that will influence design

---
[1]The Pennsylvania State University, University Park, Pennsylvania USA. Correspondence to: Ben Wortman <bvw5146@psu.edu>.

*Accepted by the ICML 2021 workshop on A Blessing in Disguise: The Prospects and Perils of Adversarial Machine Learning.* Copyright 2021 by the author(s).

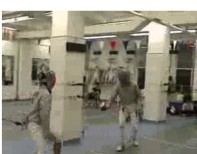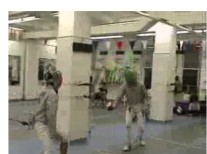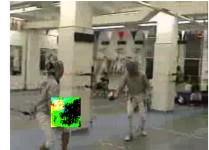

*Figure 1.* By adjusting the alpha value of the patch, we can hide it within the image. From left to right we have an alpha of 0 (no patch), 0.1, and 1 (fully opaque).

decisions. This is especially true considering (Athalye et al., 2017) has demonstrated the effectiveness of 3D printed adversarial attacks. Additionally, if it can be shown that these attacks are effective for low alpha values (transparency) such as in Figure 1, then real-world attacks become more feasible. Instead of having to develop adversarial patches for each object we apply the patch to, since trained to be invariant to the background then implementing this attack on a wide variety of objects is as simple as applying a transparent film to it. In this pilot study we show our method for carrying out such an attack and demonstrate its effectiveness.

The primary contributions of our paper include: A new hidden adversarial patch attack that is invariant to background appearance. Given its generalizability to different backgrounds, it reduces the computational cost of implementing this attack on a variety of objects in real world situations. This attack is inconspicuous to human observers.

## 2. Related Work

In this section we cover related work in disguised adversarial attacks as well as previous attacks on optical flow.

### 2.1. Disguised Adversarial Attacks

In a 2016 paper, the researchers (Kurakin et al., 2017) first demonstrated how adversarial images can be effective in real world attacks. After training adversarial patches for their image classifier, they then printed out these patches and demonstrated how these attacks can be realized in the real world. They even showed how these attacks can be disguised by constraining to optimization to fit a certain template such as peace stickers. Taking this one step further

(Sharif et al., 2016) performed a real world attack on facial recognition systems through the use of eyeglasses and other accessories. The researchers (Athalye et al., 2017) subsequently built upon these previous works by developing a new training method incorporating various transformations during training to generate attacks robust to changes in view. By doing so they were able to 3D print an adversarial turtle that tricked their object detector into thinking it was a rifle for all viewing angles.

Compared with our method, each of these attacks requires training for the individual examples. An advantage in our approach is the ability for our patches to be discretely applied to a wide variety of objects and backgrounds. This reduces training time and increases the viability of these types of attacks.

### 2.2. Attacks on Optical Flow

Recently there has been a move toward deep learning optical flow predictors over classical methods as they can provide results in near time making then ideal for applications such as automated driving. FlowNet (Ilg et al., 2017) was the first of these deep learning models, but more recently methods that combine classical and deep learning methods such as SpyNet (Ranjan & Black, 2016) and PWC-Net (Sun et al., 2018) have achieved state of the art results.

A paper by the the researchers (Sevilla-Lara et al., 2017) suggest optical flow is useful in video classification as it is invariant to appearance. Additionally, they found that the ability for optical flow to identify small movements as well as its accuracy at boundaries were both highly correlated with performance for action recognition tasks. This suggests that attacks targeted at optical flow will how downstream effects on video classifiers.

The researchers (Inkawhich et al., 2018) demonstrated this in their 2018 paper titled Adversarial Attacks for Optical Flow-Based Action Recognition Classifiers. The author demonstrates an attack on two-stream action recognition classifiers that use optical flow as an input. They generate adversarial examples targeting optical flow and show its effectiveness at misclassifying video clips from the UCF-101 dataset for both white box and black box scenarios.

In a 2019 paper, the researchers (Ranjan et al., 2019) built upon previous work by developing a white box patch attack targeting optical flow. They demonstrated the effectiveness of this attack on several deep learning optical flow models in a virtual environment before then implement this attack in real world situations by printing the patch and measuring how it disrupts optical flow.

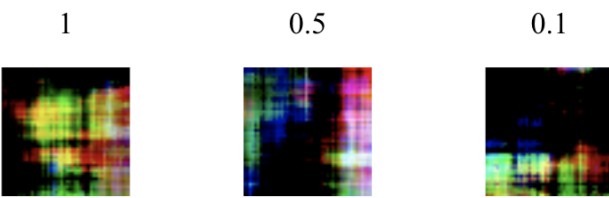

1       0.5       0.1

*Figure 2.* The patches generated during training for each alpha value.

## 3. Methodology

The full model was implemented in python and used Pytorch as the backbone for training and broadly follows the process outlined by (Ranjan et al., 2019). For our training data we used videos from the UCF-101 dataset (Soomro et al., 2012). For each video 2 consecutive frames were selected. We then use a flownet $F$ to generate a pseudo ground truth $(u, v)$ for the image sequence. SpyNet (Ranjan & Black, 2016) was selected as the flownet model for this white box attack because of its hybrid spatial pyramid deep learning structure which makes it particularly robust to adversarial examples.

Once a pseudo ground truth is established, the patch $\hat{p}$ is then multiplied by its alpha value and applied to the image sequence in a random location before the optical flow is once again calculated using SpyNet. We label the function used to insert the patch as $H(u, v, l, \alpha)$ where $\alpha$ is the transparency value and $l$ is the random location in the image sequence. The subsequent adversarial flow is $(\tilde{u}, \tilde{v})$. We then initialize our patches to cover 4.0% of the image which is comparable to the largest patch used in (Ranjan et al., 2019). Following this, loss is then calculated as the cosine similarity between the original flow values and the adversarial flow values. In summary we solve the following:

$$\hat{p} = \arg\min_{p} \frac{(u, v) \cdot (\tilde{u}, \tilde{v})}{\|(u, v)\| \cdot \|(\tilde{u}, \tilde{v})\|} \qquad (1)$$

Where

$$(u, v) = F(u, v)$$
$$(\tilde{u}, \tilde{v}) = F(H(u, v, l, \alpha))$$

Gradient descent is used to find the gradients of each pixel in the patch that minimizes this loss since a lower cosine value represents more dissimilar flows. The patch pixel values are then updated with the gradients. Since the gradients are several orders of magnitude lower than the actual pixel values, the learning rate was set to 100 in order to speed up training and ensure the pixel values update. These values are clamped so as to limit each pixel update to 2 units of change for any given iteration. This process was repeated

| Alpha | Training EPE Change | No Flow EPE Change |
|:-----:|:-------------------:|:------------------:|
| 0.1 | 61.7% | 0.42% |
| 0.5 | 13.1% | 0.51% |
| 1 | 8.3% | 0.35% |

*Table 1.* The relative change in EPE for each alpha value during both training and the zero flow test.

for alpha values of 0.5 and 0.1 to assess how this impacts performance compared to the baseline which was assigned an alpha value of 1 meaning totally opaque.

## 4. Results

To ensure the patches were training properly, we tracked the cosine similarity loss and end point error (EPE)[1] for both the training and validation data. We then selected the patch for each experiment that maximized the EPE. These patches can be visualized in Figure 2. We compare the relative difference between original and adversarial EPE for each test in Table 1. Our results show that after just 250 epochs of training each of the patches was able to increase the EPE on the validation set. Although based on this metric it appears that the lowest alpha value performed the best, the magnitude of this change isn't meaningful since in our analysis the patches remain stationary across the image sequence whereas there is significant movement in the background which will increase error for low alpha values. For this experiment we only considered whether this value increased in order to determine if it converged. In order to perform a direct comparison we conducted a zero flow test for each alpha value.

A zero flow test compares optical flow predictions between two pairs of identical images: raw and adversarial. First the flow is predicted for the raw images in order to generate the ground truth[2]. From here, the adversarial patch is then applied and it is once again passed through the flownet. The difference between these two flow fields give us an indication of how much the adversarial patch influences the predicted flow. This was measured by taking the mean EPE across the 500 validation samples, and then visualized by converting the flow field to RGB as seen in Figure 3.

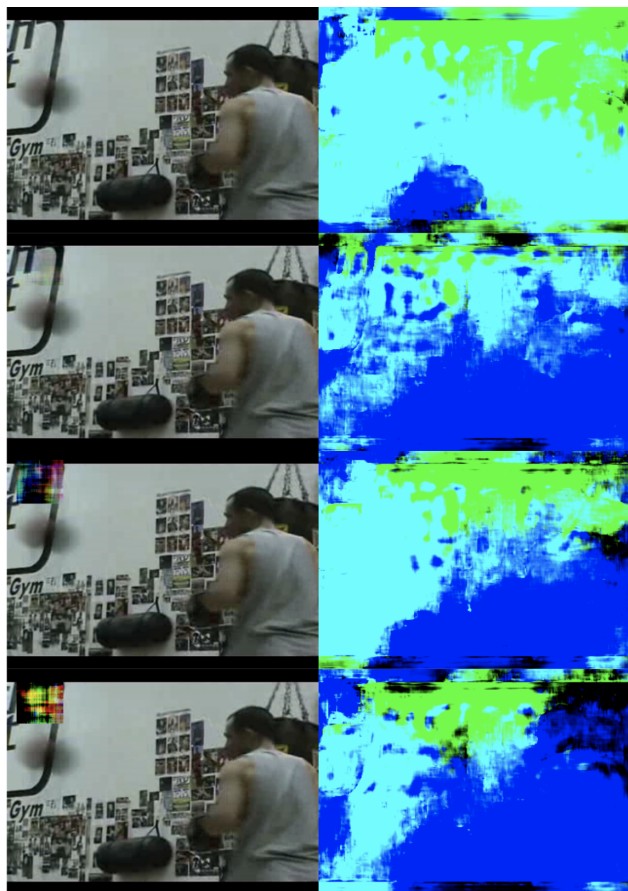

*Figure 3.* The RGB results from the zero flow test demonstrate the effect our trained patches had on optical flow. On the left side we have the raw images with the patch embedded in the top left for alpha values of 0, 0.1, 0.5, and 1. On the right we can see how the artifacts present in optical flow increase with the patch's alpha.

## 5. Discussion

We have demonstrated how hidden patch attacks can be effectively leveraged to interupt optical flow predictions in white box scenarios. Given the simplicity of this method, it can be generalized to other scenarios such black box attacks or to generate adversarial examples in image classification tasks.

As a pilot study, our approach has several limitations. First, since this experiment was run with only 2000 training samples, all models had some trouble generalizing to the validation set during training. This issue can be overcome by training on a larger more diverse training set. In addition to this, for low alpha (Transparency) values our model becomes harder to train with it often diverging. This can be overcome by reducing the learning rate and increasing the number of epochs. Finally due to the limited amount of

---

[1] A popular metric in optical flow evaluation

[2] Although the EPE should be 0 for identical images, DNN flownets have some inherent error so this is required to produce a more accurate baseline.

data, we decided to limit the transformations to randomly insterting the patch into the image. However, since previous work has already demonstrated how training with rotations and resizing can lead to more robust examples, we would expect this would apply to our method as well.

## 6. Conclusion

This hidden patch attack allows for the training of inconspicuous patches that can be applied to any background in order to disrupt optical flow. This improves their deniability and feasibility in real world attacks. Additionally, this attack is generalizable beyond optical flow to other patch attacks such as those on image classification models. Quantitative and qualitative results from the zero flow test suggest that these methods are able to disrupt optical flow and have the potential to be implemented in real world situations.

## Acknowledgements

We wish to thank Dr. Ting Wang (The Pennsylvania State University) for his valuable feedback and encouragement.

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
