# OpenReview forum: "Hidden Patch Attacks for Optical Flow"
_ICML.cc/2021/Workshop/AML — ICML 2021 Workshop AML Poster_

### Official Review · Reviewer_PszG · 2021-06-19
**The use of adversarial attack in optical flow is interesting and meaningful. However, the attack method is common and of less novelty.**

**Rating:** Reject
**Confidence:** 4

**Review:**

Strength:
1)	Easy to follow.
2)	The use of adversarial attack in optical flow task is interesting and meaningful.

Weakness:
1) There are many writing mistakes in this paper. For example, in Eq.1, the use of vertical bars is confusing. And where is Table 4 In Line130, and where is Figure 4 in Line152? Also, the patch in Figure 3 is not ‘in the top right corner’ of the image.
2) The authors just use adversarial attack methods in optical flow attack. There is less novelty in model, loss design and optimization.

---

### Decision · Program_Chairs · 2021-06-21

**Decision:**

Accept (Poster)

**Comment:**

This paper is easy to follow and studies an interesting task. The authors can further address the reviewer's comments.